# Upper Limb Strikes Reactive Forces in Mix Martial Art Athletes during Ground and Pound Tactics

**DOI:** 10.3390/ijerph17217782

**Published:** 2020-10-24

**Authors:** Vaclav Beranek, Petr Stastny, Vit Novacek, Petr Votapek, Josef Formanek

**Affiliations:** 1Department of Rehabilitation Fields, Faculty of Health Care Studies, University of West Bohemia, 30100 Pilsen, Czech Republic; 2Department of Sport Games, Faculty of Physical Education and Sport, Charles University, 16252 Prague, Czech Republic; 3Biomechanical Human Body Models, New Technologies—Research Centre, University of West Bohemia, 30100 Pilsen, Czech Republic; vnovacek@ntc.zcu.cz; 4Department of Machine Design, Faculty of Mechanical Engineering, University of West Bohemia, 30100 Pilsen, Czech Republic; pvotapek@kks.zcu.cz (P.V.); formanek@kks.zcu.cz (J.F.)

**Keywords:** mixed martial arts, system of self-defense, straight punch, palm strike, elbow strike, ground striking, head injuries

## Abstract

Athletes of mixed martial arts use a ground and pound strategy with the strikes in the dominant ground position. The aim of this study was to compare the average peak force (F_peak_) among three punches and to estimate the probability of achieving a skull bone fracture force of 5.1 kN for each type of strike in male and female athletes. A total of 60 males and 31 females (26 ± 8 years, 75 ± 20 kg, 177 ± 11 cm) practicing professional self-defense at the advanced and professional levels performed 15 strikes on a force plate. The analyses of 1360 trials showed significant differences among the strikes F_peak_ in females (*p* < 0.01) and males (*p* < 0.01). Straight punches had lower F_peak_ than palm strikes and elbow strikes in both genders, and palm strikes had higher F_peak_ than elbow strikes in females. No difference was observed between palm strikes and elbow strikes in males (*p* = 0.09). The ground and pound strikes resulted in higher impacts than previously reported strikes in the standing position. Male athletes can deliver a F_peak_ above 5.1 kN with a probability of 36% with elbow and palm strikes. Such forces can cause head injury; therefore, the use of these strikes in competition should be carefully considered.

## 1. Introduction

Mixed martial arts (MMA) is surging in popularity worldwide [1] as a modern, full-contact sport discipline, where competitors utilize different styles of martial arts and combative sports. Because of this complexity, combat athletes use several kinds of tactics, where they fight out of standing on the ground or in the standing using various strikes and other movement actions. One of the combat strategies is called “ground and pound”, where one athlete wants to obtain superior positioning over the opponent to execute strikes. Ground striking is a frequently used way of fighting when the dominant opponent sits on an athlete (mount) and uses upper limb strikes, such as a clenched fist, an open palm or an elbow [2]. Generally, the target is most often the opponent’s head because hitting the head area is a determining factor for success in MMA [3]. MMA athletes can use almost any strike at its highest intensity [4], but there is currently a lack of evidence-based comparisons of different strikes that can be used in a position on the “mount”.

Some types of strikes performed in standing stance have been compared in values of peak (F_peak_) and mean forces (F_mean_), where strikes with a clenched fist (straight punch) have been reported being higher in F_mean_ [5,6] and F_peak_ [6,7,8,9,10,11] than strikes with the open palm (palm strike) [6,12,13,14]. Conversely, the palm strikes have been reported for more effective force transferring to an object than straight punches [15]. Moreover, F_peak_ and F_mean_ may vary between genders [16,17] and level of experience [13,18,19]. In addition to a general comparison of strikes in reactive forces, it is also possible to estimate the strike potential and probability of causing bone fractures. Compilation of previous studies have reported that strikes exceeding 5.1 kN of F_peak_ would cause a skull bones fracture [6] and therefore, 5.1 kN has been defined as skull bone fracture force. This 5.1 kN threshold, represents the average load tolerance limit for bones of the skull, with the exception of the occipital region with higher load tolerance, where exact values are summarized in Table A1.

The usefulness of a strike might be represented by its velocity impact or other biomechanical advantage; the winners in elite boxing demonstrated higher strike impact represented by F_peak_ and F_mean_ than did their losing peers [8], and F_mean_ itself has been identified as one of the determinants of winning a boxing match [20]. Other studies have reported that F_mean_ is more important for success in combat sports than speed and accuracy of the strike [8,21,22]. This knowledge from boxing demonstrates that comparisons of F_mean_ and F_peak_ among different strikes is beneficial for individual combat tactics, e.g., in MMA combat where injury patterns are similar to those in professional boxing [23]. Moreover, the MMA rules allow the elbow strikes (elbow strike) for which force values reports lack in current literature.

Since ground and pound strikes have an important role in MMA tactics and might have high potential to cause injury, the aim of this study was to compare F_peak_ among kneeling straight punches with a clenched fist, palm strikes and elbow strikes in conditions close to the regular fight competition conditions in advanced male and female athletes. Furthermore, our aim also included a comparison between genders, comparison of our results with previously reported values of strikes performed in the standing stance position, calculation of the relation of F_peak_ with basic anthropometrics and calculation of the probability of achieving a skull bone fracture force of 5.1 kN for each type of strike. In the context of these aims, we hypothesized that straight punches with a clenched fist would reach higher F_peak_ values than palm strikes and elbow strikes. Another hypothesis was that the F_peak_ of ground strikes would be higher than those reported in the standing stance position, men would have higher F_peak_ than woman in each strike, and that the probability of skull bone fracture would be above 30% for all experimental strikes and in both genders.

## 2. Materials and Methods

### 2.1. Experimental Approach to the Problem

This cross-sectional study was performed during one familiarization session and one testing session separated by 48 h, where both sessions had the same schedule. The independent variables were the types of the strikes which were compared in dependent variable of F_peak_. General warm-up consisted of 10 min of jogging and stretching with supervised bodyweight exercises followed by specific warm-up involving 15 strikes of variable intensity on the measuring device. Then, the athletes received a detailed explanation of how to strike during a measurement. The participants performed 5 straight punches with a clenched fist, 5 palm strikes (straight strikes with an open palm) and 5 elbow strikes (strikes using elbow olecranon) in a randomized order.

### 2.2. Participants

A total of 60 males and 31 females (*n* = 91, 26 ± 8 years of age, 75 ± 20 kg of body weight, 1.77 ± 11 m of body height) who were practicing professional self-defense at advanced or professional levels of experience participated in the study (Table 1). At the time of testing, all participants were older than 18 years, had no injuries or other medical restrictions and signed informed consent about the purpose and contend of the study. The study protocol was approved by the local ethical committee at the Faculty of Physical Education and Sport, Charles University, Prague, Czech Republic (No. 267/2019) and was in accordance with the Declaration of Helsinki (2013).

### 2.3. Procedures of Striking Action

All data collection was carried out in a biomechanical laboratory by the same investigator and at the same hours in both sessions. Each athlete delivered 15 strikes to the force plate that was horizontally oriented with the longer side in front of the athlete. The strikes were performed from a kneeling position with a 15-s rest interval between strikes and a 5-min break between strike types.

The participants were tested with for their preferred dominant hand strikes. After adjustment the impact area of the force plate, they were asked to perform all strikes with the maximal energy that they felt comfortable with. The athletes executed an approximately perpendicular straight punch with the phalanges of the clenched fist and a palm strike in straight direction with the metacarpal area of an open palm. During the elbow strike, the athlete performed a perpendicular strike using the olecranon as the striking surface (Figure 1).

The athletes performed strikes using the specified starting position for each trial so that the results were directly comparable. The starting position of the striking hand was in contact with the lower jaw (standard defense cover). To avoid the nonstandard bending (above 40° hip flexion) of the athlete’s body above the plate before the strike, the athlete had to maintain a vertical distance from the plate to the length of the stretched upper limb in each attempt. The athletes did not touch any part of the body of the measuring device during the experiment, and the distance between the athlete’s knees from the edge of the plate was 10 cm. If the participants did not comply the specified measurement protocol, the trial was not recorded.

### 2.4. Instrumentation and Data Acquisition

All strikes were performed into a force plate (Kistler 9286B, Kistler Inc instrumente, GmbH, Winterthur, Switzerland) mounted to the floor with the participants kneeling on a raised surface next to the force plate. The apparatus consisted of a force plate with a built-in charge amplifier, Type 1758A connection cable, Type 5695B DAQ system and BioWare Type 2812A software. The sampling frequency was 10,000 Hz. The normalized weight was set to 80 kg. The measured force threshold was set to 10 N, and the default setting of the force plate for axis “z” (depth) was modified to account for the foam height from the default value of −22 mm to −40 mm. Each file contained the time, three force components Fx, Fy, Fz, and total force Ft. F_peak_ was defined as the maximum reached force during the recording interval for each trial.

The measurement plate was covered with densely dimensioned polyethylene 1.8 cm thick (Tatami Trocellen) covering because the athlete did not use any protective equipment during the measurement. Hardness was determined using a durometer (type A, DIN 53505; ASTM D 2240; ISO 7619) with different hardnesses for both sides: 20 and 35. Dynamic attenuation (accelerometer, amplifier, SW Spurt) was measured from its own frequency and relative attenuation measurements: 13.7%. The impact attenuation, which is dependent on the compressibility of the damping foam, was measured (load cell). The plate was deformed by a force of 500 N on an area of 20 cm2 from a width of 18 mm to a width of 1 mm, where further compression was no longer possible and the plate attenuation was negligible. The attenuation foam influences the measured force values with a dynamic attenuation of 20% and an impact attenuation of 500 N.

### 2.5. Statistical Analysis

The data were analyzed in MATLAB^®^ R2019b (The Math Works, Inc., Natick, MA, USA), including Statistics and Machine Learning Toolbox™ used for statistical analysis, where the significance level for all statistical tests was set to 5%. Microsoft^®^ Office Excel 2010 (Microsoft, Redmond, WA, USA) was used for the descriptive statistics and correlations between anthropometric and peak force values.

For comparisons among the strikes, the best and the worst F_peak_ performance was removed for each subject, and the mean from remaining trials were used for statistical analysis. Regarding comparisons with the literature and the probability assessment, all F_peak_ values were conserved for the statistical analysis.

The data normality for each analyzed subgroup was assessed using the Lilliefors test and one-sample Kolmogorov–Smirnov test. The F_peak_ values between the strikes for each gender were separately compared by Kruskal–Wallis test with Tukey’s honestly significant difference post hoc test and were considered significant at *p* < 0.01. Additionally, a two-sample left-tailed Wilcoxon rank-sum test was used to compare female and male results for each of the three techniques. It tested the alternative hypothesis that the F_peak_ median of the strike technique in females was lower than the F_peak_ median of the strike technique in males.

A one-sided z-test was used to compare our values to those reported in strikes in the standing stance position from previous studies.

To be able to measure if a skull bone fracture force has been achieved, the probability *P* of achieving a threshold peak force $\bar{F}$ was calculated by the Rayleigh cumulative distribution function for each strike and gender as follows:$$P=1-\overset{\bar{F}}{\underset{0}{\int }}\frac{x}{b^{2}}\exp\left(-\frac{x^{2}}{2b^{2}}\right)dx$$
where *b* is the corresponding Rayleigh scale parameter and x is the peak force data for a given data subset.

## 3. Results

The acquisition procedure resulted in 1360 successfully processed values of strikes, which were categorized for subsequent analyses (Table 2).

Five strikes were removed due to improper execution, and eleven trials were discarded due to excessively high initial total force (>30 N) exceeding the established collection threshold. The anthropometric values of body height and weight did not correlate with F_peak_ values (Figure 2). The data normality was rejected for all subsets (all at *p* < 0.01); instead, the data met the characteristics of the Rayleigh distribution. These results and basic characteristics of strikes are summarized in Table 2.

The Kruskal–Wallis test indicated significant differences among the strikes for F_peak_ values in females (H = 92.85, *p* < 0.01) and males (H = 49.42, *p* < 0.01), where the post hoc tests showed that straight punches had lower F_peak_ values than palm strikes and elbow strikes in both genders (Figure 3), and palm strikes had higher F_peak_ values than elbow strikes in females. No difference was observed between palm strike and elbow strike in males (*p* = 0.09) (Figure 3).

The two-sample left-tailed Wilcoxon rank-sum test rejected the null hypothesis for all three techniques (*p* < 0.01) and confirmed that female F_peak_ median values were lower than male F_peak_ median values for all strikes.

The comparison of our strikes to the values reported for strikes in the standing stance position was possible only for straight punches and palm strikes. One-sided z-tests confirmed that our F_peak_ values did not differ from values (3.4 ± 0.8 kN) reported by Walilko et al. [5] and were greater than the F_peak_ values reported in other literature for both straight punches and palm strikes (Figure 4 and Table A1 and Appendix A). Despite nonnormality of the collected data, the z-test was used for comparison because only means and standard deviations had been reported, thereby preventing the use of a nonparametric statistical test.

The threshold F_peak_ of 5.1 kN was selected with respect to the reported strength of various cranial bones (Appendix B). The comparison was performed for all techniques and for male, female, and all subjects. When considering all participants, there were 10.2%, 26.2%, and 27.4% probabilities of exceeding the selected force threshold with the straight punch, elbow strike and palm strikes, respectively. These are the probabilities that most of the cranial bones would suffer from serious injuries. When stratifying the data according to gender, these probabilities increased to 18.3%, 36.3%, and 36.1% with straight punches, elbow strikes and palm strikes, respectively, in the male group. In the female group, these percentages were much lower, with 0.1%, 2.5%, and 6.0% with straight punches, elbow strikes and palm strikes, respectively. These results are summarized in Figure 5 and Figure 6.

## 4. Discussion

Our results did not confirm our hypotheses because palm strikes and elbow strikes have higher F_peak_ values than straight punches in men, and because palm strikes had higher F_peak_ values than straight punches and elbow strikes in women. On the other hand, our reported results are in accordance with the results of one previous study [15], where palm strikes in a standing stance proved to have the greatest average magnitude compared to a straight punch in a standing position. One explanation for this result is that the force and energy transfers through the forearm more efficiently than through the metacarpals, and palm strikes would be a better way to transfer force to the target. Regarding elbow strikes, it was assumed that athletes use a significantly greater weight of his or her body to deliver the elbow strike because the athlete does not reach the target area by simply extending the upper limb. The reason is that the olecranon impact area is farther from the target than the hand impact area; therefore, athletes must always deliver a strike along with movements of the whole trunk. A longer distance can increase the force of the strike [13,15,24]. In comparison to strikes in standing stance, the ground and pound strikes reach a significantly higher average F_peak_. Therefore, we can conclude that different body positions during ground striking positively change how the attacker uses the substantial weight of his or her body to support the punch. The horizontal impact area on the ground provides biomechanical advantages, in that the upper limb is located under the mass of the upper torso and is moving the entire time while this movement is also enhanced by gravitational acceleration. Moreover, the ground and pound scenario decrease the submissive opponent head potential to move in free space to diminish strike impact. In worst case, the submissive opponent head might face the full impact absorption if in contact with the surface on strike impact.

The high reported kneeling strike forces, in particular, the greater than 36% probability for achieving a 5.1 kN F_peak_ value with palm strikes and elbow strikes in men, is alarming from a legal point of view. The value of 5.1 kN represents the average limit for skull bone load tolerance for the front and side regions (Appendix B). Upon reaching this value as a result of a strike, a high-risk fracture can be assumed for athletes during fight competitions, even in self-defense situations. Moreover, those results were not correlated with the participant’s body weight and body height, and therefore, there are high risks of injury following kneeling palm and elbow strikes that might appear in any weight category. Although previous studies have agreed that there is lack of correlation of the impact or effective mass of a upper limb strike or kick with the height or weight category [8,13,25], this consequence in the kneeling position is surprising and may be related to the advanced combat skills of the selected participants. On the other hand, this is not consistent with the results of recent studies that confirmed a correlation between impact force and weight category in boxing competitions [5].

The results of our analysis confirmed the hypothesis that palm strikes and elbow strikes reach significantly higher F_peak_ values than straight punches with a clenched fist in women. The average F_peak_ values with palm strikes and elbow strikes were approximately 1000 N higher than those by straight punches. In the complete dataset that included extreme values, the highest strike force measured was for the elbow strike (13,188.2 N) and next for the palm strike (9804.2 N). However, the elbow strike showed a lower median than the palm strike. The reason may be that an elbow strike among the ground and pound tactics is not so familiar, unlike the hand strikes; this was similar to what was observed in a standing stance where the impact surface of the elbow may be subjectively more sensitive than the impact surface of the open palm, and thus, the athletes did not perform maximum strikes by elbow in all trials. Furthermore, in the case of an elbow strike, the athlete had to make a longer movement of the whole torso/upper part of the body, which makes it more difficult to perform, and the longer movement also means that it is more difficult to control the falling limb. As a result, the subjects could intentionally perform the movement with less force, and the strike area was also vertically lowered to the level of the measuring platform; however, the opponent’s head was at a higher height during testing and training. The dominance of palm strikes and elbow strikes was also confirmed by the conclusions of our probability results, where a significantly higher probability of reaching a peak force of 5.1 kN occurred with elbow strikes and palm strikes than with straight punches. In the overall results across genders, only 10.2% of strikes reached the level of 5.1 kN for a straight punch with a clenched fist as opposed to elbow (26.2%) and palm (27.4%) strikes.

Significantly higher F_peak_ values for strikes in the combat style using “ground striking” compared to strikes in a standing stance confirmed the assumption that the position of body while ground striking positively affects the impact force of the strikes. Data from the literature allowed us to compare 7 sets of results with the straight punch and 4 sets of results with the palm strike (Table A2). In addition to the Walilko et al. [5] study of Olympic boxers, straight punches and palm strikes while ground striking achieved significantly higher average forces than straight punches and palm strikes in a standing. Only data from the literature that reported the F_peak_ with standard deviations could be included in the comparisons, while other studies were excluded. The included studies from the literature reported a lower number of subjects in total (up to 48 subjects) than the current study (91 subjects). Likewise, the literature studies reported data only from those with professional-level experience, while the current study reported data from 10 professionals and 81 advanced level athletes, which supports the evidence for the observed results regarding differences between ground and standing stance striking.

The results of the measurements also showed that the F_peak_ values for three types of punches significantly exceeded the load tolerance of bone tissue. Male athletes with an average weight of 75 kg and height 1.7 m can deliver sufficient F_peak_ values to break all bones of the front and side of the skull, with the exception of the occipital bone, with a probability of 36% for palm strikes and elbow strikes. Strikes on the spine and back of the head are not allowed in MMA competition; therefore, only the facial and lateral parietal parts of the head were rated. In MMA, all three punches are allowed during ground striking. Unlike fighting in a standing stance, a submissive opponent is at a disadvantage because this individual cannot mitigate the impact forces of the hit by changing distance. For this reason, it is possible to assume even greater likelihood of injuries for the combat athlete in a competition. On the other hand, our study provided the optimal time and distance conditions to produce the highest F_peak_ values, and these conditions do not necessarily happen during competition, especially due to the sudden timing of the strikes to hit opponents.

Generally, there has been significant scrutiny from medical associations regarding the high rates of trauma in MMA [26], where aggressive techniques present substantial injury risk [27]. MMA fighters were significantly more likely to experience injury (typically contusion/bruising) compared to boxers [27]. Because blows directly to the head are an effective way to achieve a win, MMA has reported even higher rates of traumatic brain injury than those assessed in American-style football, ice hockey or other contact sports [28,29], and the injury incidence in MMA appears to be greater than in most, if not all, other popular and commonly practiced combat sports [23]. In contrast to these findings, Curran-Sills and Abedin [30] reported that MMA does not confer the same exposure to concussion over a 10-year period as seen in other popular sports (e.g., ice hockey, American football, rugby union), but it is important to avoid this simplified derivation because of methodological differences across the studies [30]. A recent study found that head injuries dominate in MMA [23,30,31,32,33]. Professional mixed martial artists (MMA fighters) as such boxers are at risk of sustaining acute head and neck injury each time they engage in practice or competition [34]. These athletes are exposed to the cumulative long-term neurocognitive sequelae of repetitive insults [35]. Brain injury arising from head trauma is a major concern in MMA because knockout (KO) or technical knockout (TKO) are frequent fight outcomes, many movement combinations are targeted to hit the head area [36], and previous studies have shown a high incidence of matches ending due to strikes to the head. Strikes to the head are the major techniques used to end a match via KO/TKO, regardless of sex and weight class [28]. Based on an examination of 440 matches from 2002 to 2014, the main cause of injuries in doctor-stoppage situations was facial injuries (90%), with 87.1% occurring after striking actions. This report also showed higher values related to striking the head in stand-up actions as well as on-the-ground actions [4]. Rates of KOs and TKOs in MMA are higher than previously reported rates in other combative and contact sports. Competition data and video records for all KOs and TKOs from numbered Ultimate Fighting Championship MMA events (n = 844) between 2006 and 2012 identified that all KOs were the result of direct impact to the head, most frequently a strike to the mandibular region (53.9%) [29]. Similarly, head injuries were the most common injuries recorded in 285 championship fights between 2016 and 2018, where head injuries were significantly associated with KOs. Additional research should be extended to head-related injuries in MMA matches, especially those associated with KOs/TKOs [37], including an examination of injury prevention policies to limit injury risk in MMA [1].

### A Limitation of the Study

A major limitation of the study is the use of only the dominant side, which we expected to be preferred by athletes and might be the main premise of future studies. The fact that participants did not wear competition hand protection was accounted for by the foam covering force plate. However, the inelastic approach to the measurement cause the impact hardness which may be a psychological barrier for the striker due to fear of injury. Therefore, it is necessary to use sufficient damping because the elasticity of the target has an influence on the measurement of the impact force [38]. Other option is the use of piezoelectric sensors as accelerometers [5,7,39], force sensors based on strain gauges [19], high-speed cameras [36,40,41], use of the dummy [42], or a combination of those techniques [14]. This article presents F_peak_ strikes comparison without the lower and upper extremes, which are typical for F_peak_ results and cause high individual co-efficient of variation (Table 2). However, the results are the same even for whole datasets including extremes. On the other hand, some results might slightly alternate if strikes comparison would be done only for highest F_peak_ values or other special selection, which is available in Appendix A.

## 5. Conclusions

Straight punches, palm strikes, and elbow strikes are effective solutions for victory in ground and pound tactics due to their high impact potential, but they should also be considered high injury-risk strikes. MMA athletes, trainers, self-defense or tactical coaches can expect a high F_peak_ (2900–4100 N) on average for both genders. No significant difference was found in the effectiveness of palm strikes and elbow strikes in men; therefore, both strikes should be preferred in men during ground and pound striking because they each have the advantage of high impact peak forces. In contrast, straight punches showed lower impact forces (to 3000 N) in both genders, and the impact force will probably be further attenuated during the match by gloves. Palm and elbow strikes also provided extreme reactive maximum peak forces (for men, 13 kN), which have a high probability of causing head injury. The elbow strike has the highest potential to reach extreme impact values and fracture skull bones.

## Figures and Tables

**Figure 1 ijerph-17-07782-f001:**
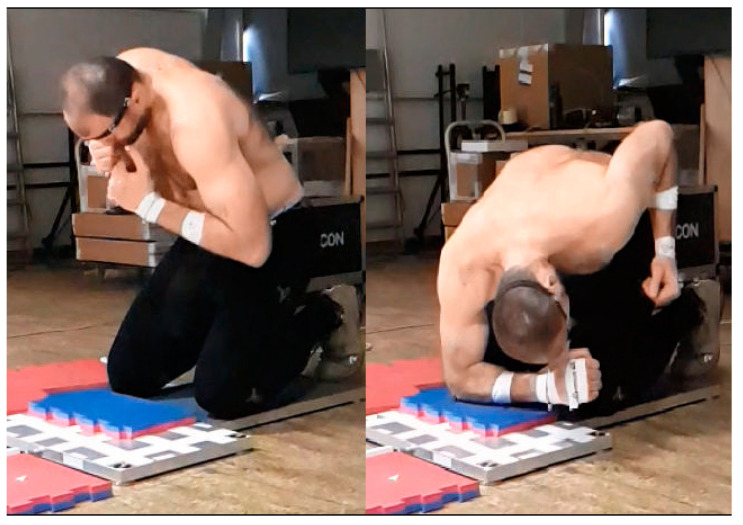
Start (left part) and finish (right part) positions for the elbow “ground” strike.

**Figure 2 ijerph-17-07782-f002:**
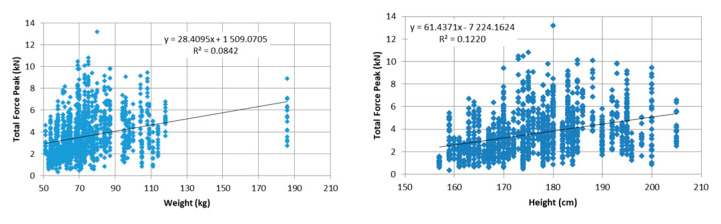
Correlation between peak force and participants weight and height.

**Figure 3 ijerph-17-07782-f003:**
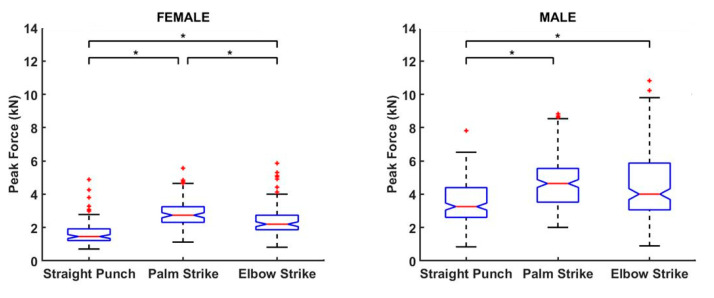
Peak force comparison of male and female strike techniques. * significantly different at *p* < 0.01.

**Figure 4 ijerph-17-07782-f004:**
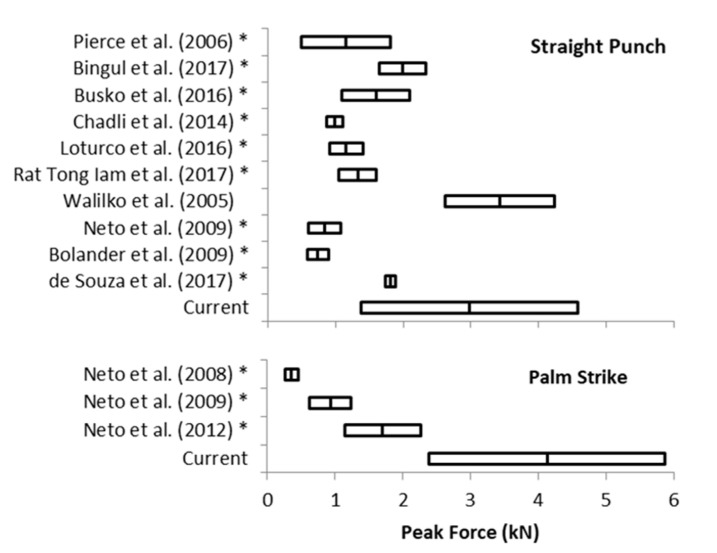
Comparison of straight punch and palm strike peak forces reported in previous studies. * significantly lower than current study. The bar present the means and standard deviations.

**Figure 5 ijerph-17-07782-f005:**
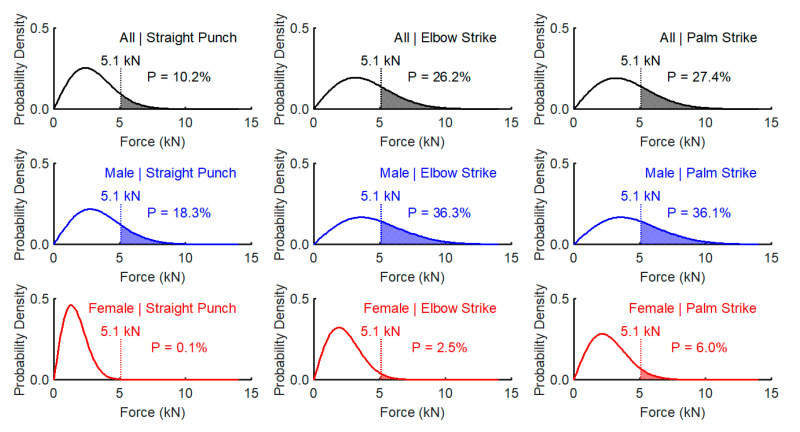
Probability of exceeding the 5.1 kN threshold force. The blue color–Male, the red color-Female.

**Figure 6 ijerph-17-07782-f006:**
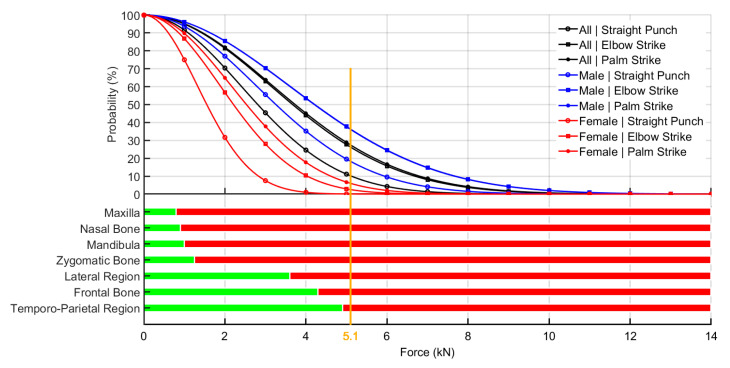
Probability of different strikes to exceed the 5.1 kN and bone tolerance. Green color–limits of allowed average load tolerance of bone fracture. Red color–over the limit of allowed average load tolerance of bone fracture.

**Table 1 ijerph-17-07782-t001:** Mean values with standard deviations of anthropometric characteristics of participants.

Sex	Experience	*n*	Age ± SD, y, [MIN, MAX]	Height ± SD, cm [MIN, MAX]	Weight ± SD, kg [MIN, MAX]
Female	Advanced	30	21 ± 1 [19, 23]	167 ± 6 [157, 179]	61 ± 7 [51, 77]
Male all		61	28 ± 9 [20, 48]	182 ± 9 [164, 205]	82 ± 20 [68, 186]
	Advanced	51	26 ± 8 [20, 45]	180 ± 7 [164, 198]	76 ± 11 [58, 105]
	Professional	10	37 ± 6 [29, 48]	195 ± 7 [175, 205]	113 ± 27 [75, 186]
All		91	26 ± 8 [19, 48]	177 ± 11 [157, 205]	75 ± 20 [51, 186]

**Table 2 ijerph-17-07782-t002:** Basic characteristics of strikes peak forces across the study subgroups (from all trials).

Gender	Technique	N	Mean [kN] (95% CI)	SD [kN]	CV [%] (95% CI)	b [kN] (95% CI)	Min [kN]	Max [kN]
Female	Straight Punch	149	1.66 (1.51; 1.82)	0.74	26.90 (22.68; 31.13)	1.29 (1.17; 1.43)	0.71	4.88
	Palm Strike	150	2.88 (2.70; 3.06)	0.87	17.55 (15.46; 19.63)	2.13 (1.93; 2.37)	1.12	5.55
	Elbow Strike	150	2.44 (2.24; 2.64)	0.96	23.24 (19.30; 27.18)	1.85 (1.68; 2.06)	0.82	5.84
Male	Straight Punch	301	3.55 (3.36; 3.74)	1.29	25.97 (22.34; 29.60)	2.67 (2.49; 2.88)	0.84	7.83
	Palm Strike	300	4.75 (4.51; 4.99)	1.61	18.85 (16.38; 21.32)	3.55 (3.30; 3.83)	2.01	8.83
	Elbow Strike	299	4.49 (4.19; 4.78)	2.02	28.15 (24.56; 31.73)	3.48 (3.24; 3.75)	0.90	10.80
Both Genders	Straight Punch	450	2.92 (2.75; 3.10)	1.45	26.28 (23.52; 29.04)	2.30 (2.17; 2.45)	0.71	7.83
	Palm Strike	450	4.12 (3.92; 4.32)	1.66	18.42 (16.65; 20.19)	3.14 (2.96; 3.34)	1.12	8.83
	Elbow Strike	449	3.80 (3.56; 4.04)	1.99	26.53 (23.80; 29.26)	3.03 (2.86; 3.23)	0.82	10.80

N = number of analyzed strikes; SD = standard deviation; CI = confidence interval; CV = mean individual co-efficient of variation; b = scale parameter of Rayleigh distribution; kN = kilonewtons; Min = minimum; Max = maximum.

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
