# Peer review of "Upper Limb Strikes Reactive Forces in Mix Martial Art Athletes during Ground and Pound Tactics"

_ijerph, 2020, doi:10.3390/ijerph17217782_

Round 1

Reviewer 1 Report

General

The authors have sought to investigate the strike forces produced by the closed fist, open palm and elbow techniques as used in MMA competitive bouts. The investigation is contemporary and should have relevance to the readership, especially with the facial injury context provided by comparison to reported forces leading to facial trauma.

Specific

Ln67-68; Can the authors please clarify why the hypothesis indicates that a palm strike would have a higher Fpeak when at Ln 45-46 the authors cite 17 articles that report higher Fpeak for a closed fist strike? What is the rationale for this hypothesis? Considering the information as presented it would be better to hypothesis that the closed fist would be higher (as consistent with previous research) and then state in the discussion that the reported data did not support the hypothesis and to then present reasons for why this inconsistency has been recorded.

Ln 96; Please delete, “….and consisted of 5 straight punches, 5 palm strikes and 5 elbow strikes in 97 randomized order by strike type….” As it is a repeat of information provided earlier at Ln79-80.

Ln 117; Comparing Figure 1 to the text , the force plate is not embedded in the floor it is mounted or positioned on the floor with the participants kneeling on a raised surface. The participant as shown has 3D retroreflective markers in what appears to be a full body mark-up, why? It would be disappointing if the authors have another separate data set from a kinematic analysis that has not been included with this manuscript.

Ln 120; This seems to be an extremely high sampling rate for this impact detection. Considering that for gait heel strike during running the sampling frequency may only be increased to 3000 Hz what is the justification for such a high rate? There is no indication of any signal treatment, was the reaction force filtered in any way, spectrum power analysis of the recording to determine up and lower bandpass filtering?

Table 2; Please include the sample size (number of punches analysed) in brackets alongside each technique. It is unclear why there is a 95% CI presented for the SD. As each participant delivered 5 strikes per technique it would aid the reader greatly to understand the variability of these strikes, thus please include the mean individual co-efficient of variation (CV) for each technique and the associated 95% CI.

Ln 195-197; The sentences is grammatically incorrect, please review and correct.

Ln 197-199; Although the result may be supporting one previous report it is in contrast to 17 other reported investigations. This needs to be reflected in the wording of the comparison and contrast to the literature.

Ln 207-211; A further discussion point to be considered is that in landing any strike while standing the target area (head) has the potential to move in free space and absorb through flexion and extension the impact of the blow. However in the ‘ground and pound’ scenario there is no further movement of the head other to absorb the full impact of the blow.

Ln 300-314; This information is not relevant and should be deleted. A limitation to the study, is the use of only the dominant side, the fact that participants did not wear competition hand protection (gloves) and whether the compressibility of the gloves was similar in any way to the foam force plate covering? The authors briefly comment on the fact that instruction during training is to aim for the back of the head with any strike, where as in this instance because a test dummy was not used the self protection element for the strike cannot be discounted. It would be interesting to understand whether those participants that performed the elbow strike first had lower relative values to their performance of the closed fist and palm strikes as they needed the time extra strikes to become comfortable with the style and depth of contact?

Ln315-324; Please move this information to the methods section as it is not a limitation to the study but a confirmation of the study data collection methodology.

Author Response

General

The authors have sought to investigate the strike forces produced by the closed fist, open palm and elbow techniques as used in MMA competitive bouts. The investigation is contemporary and should have relevance to the readership, especially with the facial injury context provided by comparison to reported forces leading to facial trauma.

Answer: Thank you for your positive comments and concerns about our article. We have revised the research question and addressed all of your comments, which sufficiently improved our manuscript.

Specific

Ln 67-68; Can the authors please clarify why the hypothesis indicates that a palm strike would have a higher Fpeak when at Ln 45-46 the authors cite 17 articles that report higher Fpeak for a closed fist strike? What is the rationale for this hypothesis? Considering the information as presented it would be better to hypothesis that the closed fist would be higher (as consistent with previous research) and then state in the discussion that the reported data did not support the hypothesis and to then present reasons for why this inconsistency has been recorded.

Answer: We agree that the there is inconsistency in the hypotheses and literature and discussion approach. Therefore, we revised the hypotheses according to your suggestion. Originally, we based our hypotheses on Empirical approach and results by Bolander 2009. However, you are right that we should stick on literature findings for the scientific introduction.

We have corrected also the statement in introduction (second paragraph) about the strikes, where we mentioned the literature inconsistency between Fpeak findings and normalized F and force transfer findings (by Bolander 2009).

Other related changes were at the beginning of discussion, where we rephrase it main results according to “new” hypotheses.

Ln 96; Please delete, “….and consisted of 5 straight punches, 5 palm strikes and 5 elbow strikes in 97 randomized order by strike type….” As it is a repeat of information provided earlier at Ln79-80.

Answer: We agree, and we deleted this information repeat.

Ln 117; Comparing Figure 1 to the text , the force plate is not embedded in the floor it is mounted or positioned on the floor with the participants kneeling on a raised surface. The participant as shown has 3D retroreflective markers in what appears to be a full body mark-up, why? It would be disappointing if the authors have another separate data set from a kinematic analysis that has not been included with this manuscript.

Answer: Sorry for this confusion, we used mounted (not embedded) version of force plate positioning as is on the figure. We corrected the text about the placement of the force plate. We agree that showing markers on figure would is not appropriate for our presented results, therefore we are providing new figure without the markers.

Ln 120; This seems to be an extremely high sampling rate for this impact detection. Considering that for gait heel strike during running the sampling frequency may only be increased to 3000 Hz what is the justification for such a high rate? There is no indication of any signal treatment, was the reaction force filtered in any way, spectrum power analysis of the recording to determine up and lower bandpass filtering?

Answer: There are two reasons for the data filtering. The first, we get inspired by sampling rate used in Bolander 2009 study, although he used different devices, however at 10kHz. Second reason was that filtering smooths out the force peaks which are of the utmost interest for our approach. Thus, no data filtering was used. Preliminary test showed that force peak duration was around 20 ms. High acquisition frequency allows to cover this short period with 200 data points to properly capture the maximum force.

We have to admit that the frequency might be lower, but the higher frequency did not discard any dataset.

Table 2; Please include the sample size (number of punches analysed) in brackets alongside each technique. It is unclear why there is a 95% CI presented for the SD. As each participant delivered 5 strikes per technique it would aid the reader greatly to understand the variability of these strikes, thus please include the mean individual co-efficient of variation (CV) for each technique and the associated 95% CI.

Answer: Thank you for this suggestion, we now added sample size

Ln 195-197; The sentences is grammatically incorrect, please review and correct.

Answer: We revised and corrected this sentence.

Ln 197-199; Although the result may be supporting one previous report it is in contrast to 17 other reported investigations. This needs to be reflected in the wording of the comparison and contrast to the literature.

Answer: We are now stating that there is only “one” study like this. Moreover, we included this study to the introduction as contrary to other studies. We also removed some studies from introduction according to other reviewer suggestion, and left only those with apparent differences.

Ln 207-211; A further discussion point to be considered is that in landing any strike while standing the target area (head) has the potential to move in free space and absorb through flexion and extension the impact of the blow. However in the ‘ground and pound’ scenario there is no further movement of the head other to absorb the full impact of the blow.

Answer: Thank you for this suggestion, we now added the point of decreased spatial opportunities of submissive opponent head, which actually makes even smoother transfer to next paragraph.

Ln 300-314; This information is not relevant and should be deleted. A limitation to the study, is the use of only the dominant side, the fact that participants did not wear competition hand protection (gloves) and whether the compressibility of the gloves was similar in any way to the foam force plate covering? The authors briefly comment on the fact that instruction during training is to aim for the back of the head with any strike, where as in this instance because a test dummy was not used the self protection element for the strike cannot be discounted. It would be interesting to understand whether those participants that performed the elbow strike first had lower relative values to their performance of the closed fist and palm strikes as they needed the time extra strikes to become comfortable with the style and depth of contact?

Answer: We were trying to show the limits of selected measurements, which we agree is redundant. Therefore, we now put only one paragraph for this section, which is mentioning the hand preference, missing protectors and other measurement approaches such as the use of dummy.

Ln315-324; Please move this information to the methods section as it is not a limitation to the study but a confirmation of the study data collection methodology.

Answer: We agree, we moved this part to the methods into the force plate set up section.

Reviewer 2 Report

Line 1/2. Please in the the title don’t use acronims.

Abstract

Line 21. Please, the first time a acronim appears it must be defined. In this case “Fpeak”.

Introduction

In the article the references, except when they are repeated, should be in increasing order.

In the introduction the first reference is number 17, followed by 45, 30, 37 ... This should be thoroughly reviewed throughout the text.

There are 17 references on line 46. This is excessive and you should select the ones that are really relevant and calming. They also appear in a disorganized order.

Line 63. At the end of athletes there is a quote that should not be.

Material and Methods

In table 1, line 92, the unit of height it’s in meters, but it’s imposible the mean could be 167, 182, 180, 195, 177 meters!!! Are yo sure. Maybe the unit could be cm? Please correct that.

Results

Line 152, Table2. Please, include the acronim “kN” in the list.

Line 179. There are two docs at the end of the line, please correct it.

Discussion

Without a doubt, this is the best section of the article, in which an in-depth review and comparison of the results obtained with previously published studies is made. They also make an interesting practical application that for scientists, athletes and coaches will be of enormous value.

The singularity of this work can be found int he interesting question posed in the hypothesis with a view to safeguarding possible injuries of MMA practitioners.

Author Response

Answer:

Dear reviewer,

Thank you for your positive evaluation and comment to improve our manuscript. We have resolved all your comments according to your suggestion.

Abstract

Line 21. Please, the first time a acronim appears it must be defined. In this case “Fpeak”.

Answer: Done, we have defined the abbreviation before first use.

Introduction

In the article the references, except when they are repeated, should be in increasing order.

Answer: Corrected, sorry for this accident.

In the introduction the first reference is number 17, followed by 45, 30, 37 ... This should be thoroughly reviewed throughout the text.

Answer: Corrected, sorry for this accident.

There are 17 references on line 46. This is excessive and you should select the ones that are really relevant and calming. They also appear in a disorganized order.

Answer: We agree and carefully revised the references, which resulted in rewording the sentence and the use of lees amount of references.

Line 63. At the end of athletes there is a quote that should not be.

Answer: We agree, therefore we deleted this quotation.

Material and Methods

In table 1, line 92, the unit of height it’s in meters, but it’s imposible the mean could be 167, 182, 180, 195, 177 meters!!! Are yo sure. Maybe the unit could be cm? Please correct that.

Answer: Thank you for catching this inaccuracy, we corrected to the cm.

Results

Line 152, Table2. Please, include the acronim “kN” in the list.

Answer: We have added the kN description and also other added parameters description.

Line 179. There are two docs at the end of the line, please correct it.

Answer: Corrected

Discussion

Without a doubt, this is the best section of the article, in which an in-depth review and comparison of the results obtained with previously published studies is made. They also make an interesting practical application that for scientists, athletes and coaches will be of enormous value.

Answer: Thank you for this comment, we now little more improved the discussion part.

The singularity of this work can be found int he interesting question posed in the hypothesis with a view to safeguarding possible injuries of MMA practitioners.

Answer: Thank you for this comment, we now little more improved the discussion part.

Round 2

Reviewer 1 Report

Ln 124; After the closed bracket insert the word 'covering' to improve the sentence grammar

Ln 161; Please confirm that the presented CV% (95%CI) is the average of the participants and not of the whole sample. The request is to present the intra-individual variation of the 5 strikes per condition as the values are very high. If the values are correct then the high variability deserves further discussion. Was the highest force strike always the 5th or was this evenly distributed? How does the analysis change if A) only the highest impact is compared between conditions, or B) the mean of strikes 2, 3, and 4 are compared to account for learning and other confounding variables?

Ln 230; Delete the word 'leaning' replace with, 'in contact with the surface on strike impact.'

Ln 321; Replace 'contend' with 'premise'

Ln 322; Replace 'compromise' with 'accounted for'

Author Response

Ln 124; After the closed bracket insert the word 'covering' to improve the sentence grammar

Answer: Done, Thank you for this improvement and again for your valuable comments and time spend to improve our manuscript.

Ln 161; Please confirm that the presented CV% (95%CI) is the average of the participants and not of the whole sample. The request is to present the intra-individual variation of the 5 strikes per condition as the values are very high. If the values are correct then the high variability deserves further discussion.

Answer: We are confirming that CV% is from the participant average typically from 5 strikes (perhaps four strikes in couple individuals). This was actually one of the reasons, why we discard the lowest and highest Fpeak in each individual and used mean from remaining trials for the Fpeak strikes comparison. For other analyses we used the full set of data. This we are clearly stating at lines 139-142.

For better clarity, we now stated in the Table 2, that those data are from 5 trials. We can claim that for reduces data set there is much lower CV´s. Since we uploaded raw dataset, interested readers can check this difference.

The reason for high CV is the parameter of the Fpeak itself (not Fmean of impulse), which is typical for random extremes (upper or lower). However, after the extremes reduction the individual variance dramatically decreases. Upon your suggestion we have put this phenomenon into limitation section along with discussion of different statistical setting.

“This article presents strikes comparison without the lower and upper extremes, which are typical for Fpeak results and cause high individual co-efficient of variation (Table 2). However, the results are the same even for whole datasets including extremes. On the other hand, some results might slightly alternate if strikes comparison would be done only for highest Fpeak values or other special selection, which is available in Supplementary material 2 - raw datasheet.“

On the other hand, we have left all extreme values for the calculation of probability, which we believe is appropriate, since the extremes are closely connected to probability of “successful / non-successful” strike.

Was the highest force strike always the 5th or was this evenly distributed?

Answer: It was evenly distributed as well as the lowest force strike. See supplementary material 2 - raw data sheet, which shows also the strike trial order. We believe that this is due to the participants familiarization and experience level.

How does the analysis change if A) only the highest impact is compared between conditions,

There is actually one difference in the women, the use of only highest Fpeak would discard the difference between elbow and palm strike. In women this is probably due to the lower level of extremes. This possibility is mentioned in limitation section now, but in our opinion, there is no reason to highlight this option due to the low probability of women group to exceed 5.1kN Fpeak.

or B) the mean of strikes 2, 3, and 4 are compared to account for learning and other confounding variables?

We actually calculated the full dataset and the dataset with extremes reduction, and those results ended up the same (only p value was higher in the full set). We additionally calculated the only middle three trials and the results were again the same. The possibility of different data selection is now stated in limitation section and any additional calculation might be done from raw dataset. We also stating at experimental approach that participants were fairly familiarized with the protocol.

Ln 230; Delete the word 'leaning' replace with, 'in contact with the surface on strike impact.'

Answer: Done

Ln 321; Replace 'contend' with 'premise'

Answer: Done

Ln 322; Replace 'compromise' with 'accounted for'

Answer: Done